# Association of the Posterior Acromion Extension with Glenoid Retroversion: A CT Study in Normal and Osteoarthritic Shoulders

**DOI:** 10.3390/jcm11020351

**Published:** 2022-01-12

**Authors:** Alexandre Terrier, Fabio Becce, Frédéric Vauclair, Alain Farron, Patrick Goetti

**Affiliations:** 1Laboratory of Biomechanical Orthopedics, Ecole Polytechnique Fédérale de Lausanne, Station 19, 1015 Lausanne, Switzerland; alexandre.terrier@epfl.ch; 2Department of Orthopedics and Traumatology, Lausanne University Hospital and University of Lausanne, Avenue Pierre-Decker 4, 1011 Lausanne, Switzerland; frederic.vauclair@chuv.ch (F.V.); alain.farron@chuv.ch (A.F.); 3Department of Diagnostic and Interventional Radiology, Lausanne University Hospital and University of Lausanne, Rue du Bugnon 46, 1011 Lausanne, Switzerland; fabio.becce@chuv.ch

**Keywords:** acromion morphology, glenoid retroversion, wear, osteoarthritis, computed tomography

## Abstract

Posterior eccentric glenoid wear is associated with higher complication rates after shoulder arthroplasty. The recently reported association between the acromion shape and glenoid retroversion in both normal and osteoarthritic shoulders remains controversial. The three-dimensional coordinates of the angulus acromialis (AA) and acromioclavicular joint were examined in the scapular coordinate system. Four acromion angles were defined from these two acromion landmarks: the acromion posterior angle (APA), acromion tilt angle (ATA), acromion length angle (ALA), and acromion axial tilt angle (AXA). Shoulder computed tomography scans of 112 normal scapulae and 125 patients with primary glenohumeral osteoarthritis were analyzed with simple and stepwise multiple linear regressions between all morphological acromion parameters and glenoid retroversion. In normal scapulae, the glenoid retroversion angle was most strongly correlated with the posterior extension of the AA (R^2^ = 0.48, *p* < 0.0001), which can be conveniently characterized by the APA. Combining the APA with the ALA and ATA helped slightly improve the correlation (R^2^ = 0.55, *p* < 0.0001), but adding the AXA did not. In osteoarthritic scapulae, a critical APA > 15 degrees was found to best identify glenoids with a critical retroversion angle > 8 degrees. The APA is more strongly associated with the glenoid retroversion angle in normal than primary osteoarthritic scapulae.

## 1. Introduction

Several measures of acromion morphology, both in the sagittal and coronal planes, have been described and associated with various shoulder disorders [1]. In recent years, much research has been directed towards the characterization of the lateral extension of the acromion. The acromion index followed by the critical shoulder angle, both described on antero-posterior shoulder radiographs, have been shown to be predictors of glenohumeral osteoarthritis (OA) and rotator cuff tendon tears [2,3]. These initial findings were supported by subsequent biomechanical studies revealing increased glenohumeral joint reaction forces with decreased lateral extension of the acromion [4,5]. However, these two anatomical parameters are unable to assess the antero-posterior imbalance of the glenohumeral joint typically found in Walch type B glenoids [6].

Over the past three years, Beeler et al. analyzed shoulder computed tomography (CT) images to improve characterization of acromion roof morphology. Compared with osteoarthritic shoulders, the acromion was more externally rotated (axial plane), more downward tilted (coronal plane), and had wider posterior coverage of the glenoid (sagittal plane) in shoulders with rotator cuff tears [7]. The same research group further found a significant difference between shoulders with concentric and eccentric primary glenohumeral OA, and concluded that a flatter acromion roof with less posterior glenoid coverage could contribute to static posterior subluxation of the humeral head and posterior glenoid wear [8]. Furthermore, Meyer et al. were able to link a decreased posterior acromion slope and increased glenoid retroversion with static posterior subluxation of the humeral head and posterior glenoid wear [9]. More recently, Beeler et al. reported that the scapula of a shoulder with dynamic and static posterior instability was characterized by an increased glenoid retroversion and an acromion that was shorter posterolaterally and higher and more horizontal in the sagittal plane [10].

Static posterior subluxation of the humeral head and posterior glenoid wear are of particular interest to shoulder surgeons, as they have been related to both early glenohumeral OA in young adults [11], and higher complication rates after shoulder arthroplasty [12]. Although Walch et al. stated that static posterior subluxation of the humeral head preceded posterior glenoid wear, with glenoid retroversion as a risk factor [11,13], there is currently no consensus regarding this chicken-or-egg debate [6,14,15]. To our knowledge and despite the recent study by Beeler et al. [10], there are no published data on the correlation between the detailed three-dimensional (3D) acromion shape and glenoid retroversion.

Therefore, our objective is to investigate the potential association between the 3D acromion shape and glenoid retroversion in both normal and osteoarthritic shoulders. This could prove to be clinically useful to better understand the etiology of eccentric glenoid wear and possibly define an anatomical parameter to predict its risk of occurrence. We first examined the presence or absence of correlation between the 3D acromion shape and glenoid retroversion in normal scapulae. Then, we tested whether the same results held true in shoulders with primary glenohumeral OA.

## 2. Materials and Methods

### 2.1. Study Population

This retrospective observational study was approved by the institutional ethics committee, with a waiver of patient informed consent (CER-VD protocol number 505-15). We considered the following two patient groups, who did not need to be matched since they were analyzed separately according to the study design and primary objective (i.e., association between 3D acromion shape and glenoid retroversion).

The normal group included trauma patients aged 18 to 40, who had undergone a whole-body CT scan covering at least one of the two scapulae in full. Exclusion criteria were any radiological sign or history in medical records of disorders of the shoulder bones and joints (OA, fracture, glenoid dysplasia, or prior surgery of the upper limb), CT signs of immature skeleton (absence of fusion between any of the scapular ossification centers [16]) or CT artifacts (motion or metal). From our institutional picture archiving and communication system, an attending musculoskeletal radiologist retrospectively reviewed 221 consecutive whole-body CT scans performed over a 6-month period, and from these 112 patients met the inclusion criteria. The main characteristics of the normal subjects (79 males and 33 females) were mean age, 28.4 years (range, 18–40); mean height, 174.4 cm (range, 150–210 cm); mean weight, 75.5 kg (range, 50–120 kg); mean body mass index, 24.7 kg/m^2^ (range, 18.6–38.1 kg/m^2^).

The pathological group consisted of patients with glenohumeral OA who had undergone a shoulder CT scan covering the entire scapula in their preoperative planning prior to shoulder arthroplasty. Patients with any traumatic injury to the shoulder girdle, malunion or nonunion, necrosis of the humeral head, or rheumatoid arthritis were excluded. Of the 334 consecutive patients eligible from 2002 to 2016, 125 with primary glenohumeral OA met the inclusion criteria. The main characteristics of the OA patients were mean age, 71.4 (range, 46–88 years); 37 males, 88 females; mean height, 165.7 cm (range, 141–186 cm); mean weight, 78.4 kg (range, 42–129 kg); mean body mass index, 28.5 kg/m^2^ (range, 17.7–43.6 kg/m^2^). According to the updated Walch classification [17], the distribution of glenoid types was: A1, n = 26; A2, n = 23; B1, n = 26; B2, n = 37; B3, n = 8; C, n = 5.

### 2.2. CT Protocols

All CT scans were performed on multidetector-row CT systems (8–256 detector rows) from the same manufacturer (GE Healthcare), with standardized data acquisition and image reconstruction settings. For normal subjects, scapular CT images were reconstructed as follows: section thickness, 1.3 mm; section interval, 0.7–1.3 mm; kernel, sharp (bone or bone plus); pixel size, 0.4–1.0 mm. For OA patients, shoulder CT images were reconstructed as follows: section thickness, 0.6–1.3 mm; section interval, 0.3–1.0 mm; kernel, sharp (bone or bone plus; GE Healthcare); pixel size, 0.3–0.6 mm.

### 2.3. Scapular Coordinate System

All CT scans were analyzed in 3D using a reliable semi-automated method providing a scapular coordinate system described in detail elsewhere [18,19]. Briefly, the medio-lateral (*z*) axis was along the scapular axis, defined by the line fitting five points placed along the supraspinatus fossa projected in the scapular plane. The scapular (i.e., ~“coronal”) plane was defined by three landmarks: the trigonum spinae (TS), the angulus inferior (AI), and the most medial of the five points defining the medio-lateral axis (Figure 1; additional illustrations on the coordinate system can be found in [19]). The postero-anterior (*x*) axis (i.e., ~“sagittal” plane) was then defined as being perpendicular to the scapular plane and medio-lateral axis. The infero-superior (*y*) axis (i.e., ~“axial” plane) was perpendicular to the other two axes. The origin of the coordinate system corresponded to the spinoglenoid notch projected on the medio-lateral scapular axis.

### 2.4. Acromion Landmarks

Two specific acromion landmarks were placed manually on its 3D surface using the same software (Amira; Thermo Fisher Scientific) and method as above [19]: the acromion angle (AA) and the most anterior point of the acromioclavicular (AC) joint (Figure 1). These two landmarks were characterized by their three coordinates in the scapular coordinate system (AAx, AAy, AAz; and ACx, ACy, ACz). Because of the expected variability in scapular size among patients, these coordinates (distances) were normalized by the scapular height, defined by the infero-superior distance between AI (AIy) and the origin of the scapular coordinate system.

### 2.5. Acromion Angles

From these two acromion landmarks, we defined four specific acromion angles (Figure 1): the acromion posterior angle (APA), the acromion tilt angle (ATA), the acromion length angle (ALA), and the acromion axial tilt angle (AXA). The APA is the angle between the infero-superior axis (Y) and the axis formed by the AA landmark and AI, projected in the plane perpendicular to the medio-lateral axis (i.e., ~“sagittal” plane). The ATA is the angle between the AA-AC segment and the *x*-axis, in the *xy* plane (i.e., ~“sagittal”). The ALA corresponds to the distance between AA and AC landmarks (in the *xy* —i.e., ~“sagittal”plane), measured as an angle from AI. Finally, the AXA is the angle between the AA–AC segment and the *x* axis in the *zx* plane (i.e., ~“axial”).

### 2.6. Glenoid Retroversion Angle

The glenoid retroversion angle (GRA) was measured in 3D from the CT scans using the same software (Amira) and method as above [19]: the angle between the medio-lateral axis (*z*) and the glenoid centerline projected in the axial plane (perpendicular to the infero-superior axis). The glenoid centerline was defined by the vector joining the center of the glenoid cavity and the center of a sphere fitting the glenoid cavity (Figure 1, right). This method has previously shown good to excellent inter- and intra-observer reliability [19]. For simplicity, we defined here retroversion as positive, and anteversion as negative. The glenoid centerline and all other 3D quantities and computations defined above were performed with Matlab (MathWorks). This script takes as input the three coordinates of all the scapular landmarks and all the points on the surface of the glenoid cavity to provide all reported measurements in the scapular coordinate system, for each case individually.

### 2.7. Statistical Analysis

Statistical analyses were performed in Matlab. First, for subjects with normal scapulae (i.e., trauma patients), we performed simple and stepwise multiple linear regressions to examine the correlation among all six acromion landmarks (AA and AC coordinates) and the GRA. We also evaluated the correlation among each of the four acromion angles (APA, ATA, ALA, and AXA) and the GRA. The quality of the regression was quantified by the root mean square error (RMSE), the coefficient of determination (R^2^), and its *p*-value. We further performed a receiver operating characteristic (ROC) curve analysis to determine which critical GRA and associated morphological acromion parameter better identified the two groups (i.e., low vs. high GRA), using the area under the curve (AUC) with the Youden index. The normality of the measurement data was verified by a Shapiro–Wilk test. As an additional analysis, differences between the normal and pathological patient groups were tested by an unpaired two-tailed Student’s *t*-test, and the effect size evaluated with Cohen’s d. We also assessed the dependence on patient demographics such as gender and age, and *p* < 0.05 was considered statistically significant.

## 3. Results

For normal scapulae, simple linear regressions showed that AAx was the acromion landmark coordinate most strongly and significantly associated with the GRA (R^2^ = 0.480, *p* < 0.0001), followed by ACx (R^2^ = 0.310, *p* < 0.0001) (Table 1). Stepwise multiple linear regressions between the six acromion landmark coordinates examined here (AAx, AAy, AAz; and ACx, ACy ACz) and the GRA confirmed the importance of AAx, and the slight improvement in the model by combining AAx with AAz and ACx (R^2^ = 0.530, *p* < 0.0001). Using this existing correlation, we were able to predict the measured GRA with an error (RMSE) of 3.6 degrees.

Of the four acromion angles examined here (APA, ATA, ALA, and AXA in Figure 1), the APA was the most strongly and significantly associated with the GRA (R^2^ = 0.482, *p* < 0.0001). Combining the APA with the ALA and ATA helped slightly improve the correlation, while adding the AXA did not. The APA was very strongly and significantly correlated with AAx (R^2^ = 1.00, *p* < 0.0001) (Appendix A). Because of this, among the 10 morphological acromion parameters (6 landmark coordinates and 4 angles), we then focused the analysis between the acromion morphology and glenoid retroversion to the APA versus GRA (Figure 2), which can be written as follows: GRA = 1.9 × APA − 25.2 (with GRA and APA in degrees). A 1-degree increase in the APA corresponded approximately to a 2-degree increase in the GRA.

For osteoarthritic scapulae, we also observed a significant positive correlation between the APA and GRA (R^2^ = 0.197, *p* < 0.0001), suggesting that a higher APA was associated with an increased GRA (Figure 2). While both the GRA and APA increased with the (alphabetical) progression of the updated Walch class, the increase in APA was not proportional to that of the GRA (Table 2, Figure 3). The ROC curve analysis predicted a critical GRA value of 8 degrees (AUC = 0.78) and a critical APA value of 15 degrees to best identify high GRA (>8 degrees) from low GRA (≤8 degrees).

Data for the six acromion landmarks, four acromion angles, and the GRA all followed a normal distribution. Of the six acromion landmarks, AAx, AAy, ACx, and ACy differed significantly between normal and osteoarthritic scapulae (*p* ≤ 0.03), but with a moderate-to-small effect size (Cohen’s d ≤ 0.64). Among the four acromion angles, only APA and ALA were significantly different between normal and osteoarthritic scapulae (*p* ≤ 0.04), but again with a moderate to small effect size (d ≤ 0.43). GRA was significantly more retroverted in osteoarthritic (11.1 degrees) than in normal scapulae (3.0 degrees) (*p* < 0.001). There were no significant differences in age between osteoarthritic scapulae having an APA above or below 15 degrees (*p* = 0.38). Regressions between GRA and APA were slightly different between males and females but remained within the 95% confidence intervals (CIs) of the entire datasets.

## 4. Discussion

Our objective was to test for correlations between scapular morphology and glenoid retroversion, both in patients with normal scapulae but also in those with primary glenohumeral OA. Two specific acromion landmarks were used and represented by their coordinates in the 3D scapular coordinate system. In normal (non-osteoarthritic) glenohumeral joints, we observed that the posterior extension of the acromion was strongly correlated with the GRA. This anatomical acromion measure was represented by the APA, a novel angular measure of the scapula. By comparison with the primary glenohumeral OA population, we found a critical APA value, which needs to be further investigated and might eventually be used as a predictive anatomical parameter or risk factor for posterior glenoid wear in osteoarthritic shoulders.

The two acromion landmarks used in this study characterized the acromion as a linear segment. Using the six coordinates of these two landmarks in the local scapular coordinate system, we tested all possible simple and multiple morphological associations between the defined acromion segment and GRA. These two scapular landmarks were carefully selected to be unaffected by osteoarthritic wear or osteophytes [18,19]. Bearing this in mind, the glenoid center was deliberately avoided, as its location can be modified by glenoid wear. The same logic was applied for the APA by selecting the inferior edge of the scapula (AI) as the third landmark. Although glenoid version is classically defined as negative when oriented posteriorly, we decided to use a positive value for the sake of simplicity. Hence, we used the term retroversion to avoid any confusion, and a positive correlation between the APA and GRA was reported here.

Our statistical analysis revealed that two of these six coordinates (AAx and ACx) were mainly associated with the GRA. These coordinates were therefore subsequently used to define angles that could be conveniently measured in daily clinical practice on sagittal-oblique reformats derived from preoperative shoulder CT scans. As highlighted by our results, these angles appeared to be reliable and easy-to-use alternatives to characterizing the acromion morphology. A previous analysis of the inter- and intra-observer variability in the positioning of scapular landmarks showed moderate to excellent reliability, with intraclass correlation coefficients ranging from 0.67 to 0.99 [19]. As expected, the correlation between the APA and GRA in normal scapulae was strong. The APA and AAx indeed had a strong linearly correlated since AAx is proportional to the trigonometric tangent function of the APA, which is highly linear between 0 and 20 degrees. This correlation was further enhanced after normalization of the AAx coordinate by the scapula height (R^2^ = 0.999 and 0.830, with and without normalization by the scapula height, respectively). For normal scapulae, the more posterior the acromion extension, the wider the GRA. This was reported by the posterior extension of the AA (AAx), and by the APA. According to our regression analysis, one degree in APA related to two additional degrees in the GRA.

When secondarily looking at osteoarthritic scapulae, the correlation between AAx and the APA was still significant but weaker than in normal scapulae (R^2^ = 0.482, *p* < 0.0001 vs. R^2^ = 0.197, *p* < 0.0001, respectively). This meant that the correlation observed in normal scapulae seemed to be disrupted in primary glenohumeral OA patients. Our hypothesis is that this might have been related to posterior glenoid wear. Previous research identified scapulae with increased glenoid retroversion or posterior glenoid wear as a risk factor for implant failure in total shoulder arthroplasty [20,21]. In addition, posterior glenoid bone loss is known to progress over a 5- to 15-year timeframe in up to 55% of patients [22]. Our research might therefore be critical for helping council patients by defining a critical APA value related to posterior glenoid wear. We first identified a critical GRA with a ROC curve analysis. Then, by using the Youden index, we determined the optimal APA cut-off value that could distinguish between scapulae above and below this critical GRA threshold.

Glenoid retroversion also correlated with ACx, but ACx strongly correlated with AAx (see correlation matrix in Appendix A), meaning that the relative AP acromion length (distance between AA and AC) had a low variability. Glenoid retroversion also negatively correlated with the lateral extension of the AA (AAz), which further partly correlated with the posterior extension (Table 1). These correlations between the two acromion landmark coordinates were also present between the four tested acromion angles, and explain why they do not all appear in the multiple correlations obtained with the stepwise multiple linear regression analysis.

It appears likely that the strong correlation observed between the acromion and glenoid in normal scapulae was determined by the end of growth [23]. While several hypotheses regarding posterior glenoid wear were raised (e.g., premorbid glenoid retroversion [24], muscular imbalance [25], and lower humeral retroversion [26]), its pathophysiology remains unknown. We might reasonably assume that the acromion affected the glenoid through the action of muscles, but this remains purely conjectural. This link might also be more deeply anchored in human evolution [27].

Normalized values of the two important acromion landmarks that are the AA and AC joint have not been previously reported. However, a wide range of acromion angles have recently been described with increasing interest in characterizing the acromion morphology. The APA presents similarities with the “posterior glenoid coverage” proposed by Beeler et al. as both are based on AAx and the medio-lateral scapular plane [7,8]. However, Beeler et al. used the glenoid center as the middle point, while we used the AI instead not only because it is not affected by glenohumeral osteoarthritis, unlike the glenoid center, but also and primarily to better correlate the APA with the posterior extension of the acromion (AAx). These two angles are thus very different, and we verified that the “posterior glenoid coverage” was not correlated with the GRA in our series of normal scapulae.

The ALA corresponded to the distance between AA and AC in the sagittal (*xy*) plane, measured as an angle using AI as the third landmark. Again, it is similar to the “overall glenoid coverage” proposed by Beeler et al. [7,8], but uses AI instead of the glenoid center as the middle point, to better correlate with the AA–AC segment (R^2^ = 0.908 vs. R^2^ = 0.093, respectively). In our series of normal scapulae, these two angles were only weakly correlated (R^2^ = 0.127), and the variability range of ALA was five times lower.

The ATA corresponds approximately to the previously defined acromion tilt [28], or 90 degrees minus the posterior acromion slope [9], or the sagittal tilt [7,8]. The ATA was 25.2 ± 8.2 (range, 5.2–46.9) degrees in our normal scapulae vs. 23.4 ± 8.7 (range, 4.5–42.5) degrees in osteoarthritic scapulae, which corresponds closely to the values in the articles referenced above.

The AXA corresponds approximately to the previously defined axial tilt angle [7,8]. The AXA was 26.1 ± 8.9 (range, 4.2–52.1) degrees in our normal scapulae vs. 29.1 ± 9.9 (range, 2.5–52.1) in osteoarthritic scapulae, which also matches the previous works mentioned above.

The main strength of the present study was the use of the 3D coordinates of two relevant acromion landmarks in a dedicated local scapular coordinate system generated from points not affected by glenoid wear that is secondary to glenohumeral osteoarthritis. This setup permitted multiple linear regression testing to identify the most significant determinants of glenoid retroversion, which was also comprehensively analyzed in 3D. A step further was taken by defining acromion angles that normalize measures to patients’ height, thereby obviating the need for subsequent data processing.

The major limitation of our method was the manual identification of the two acromion landmarks, the effect of which was minimized by the good reliability in the positioning of scapular landmarks by a single experienced human observer [19]. This could be improved by using sophisticated fully automatic landmark detection methods [29]. However, even with such automated methods, the AC might still be affected by osteoarthritis, conversely to all other anatomical landmarks used to characterize the acromion morphology, the angles, and the local (i.e., scapular) coordinate system. Nevertheless, the AC is not related to the APA or the scapular coordinate system, and its variability caused by OA is supposedly weak since we found no significant difference when comparing the normal and osteoarthritic datasets. Second, patient characteristics differed between normal and osteoarthritic scapulae, as per the study design and primary objective (association between 3D acromion shape and glenoid retroversion), and considering that trauma is more common in males and shoulder OA in females. However, our aim was to assess potential morphological associations separately, first in normal and then osteoarthritic scapulae. We verified that the same correlations held true in males and females, with variations within the 95% CIs. Regarding aging, we further checked that this demographic parameter did not affect the critical APA value reported here. Finally, although trauma patients with normal scapulae and patients with glenohumeral OA were not scanned with the same CT protocols, the differences in the reconstructed geometric volumes were small (with slightly smaller voxels for OA than trauma patients) and had no impact on the positioning of scapular landmarks.

## 5. Conclusions

The strong correlation observed here between the posterior acromion extension, in particular the APA, and glenoid retroversion in normal scapulae suggests that the APA might be used as a predictive anatomical parameter or risk factor for the development and progression of primary glenohumeral osteoarthritis associated with posterior glenoid wear. However, the identified critical APA value at 15 degrees should now be further investigated in larger patient series with osteoarthritic scapulae/glenoids, and if possible elderly controls without any sign of glenohumeral OA. Moreover, long-term clinical studies should evaluate the impact of the APA on clinical function and surgical revision rates after shoulder arthroplasty. Finally, we should also examine whether the APA is associated with other shoulder disorders or specific treatment outcomes.

## Figures and Tables

**Figure 1 jcm-11-00351-f001:**
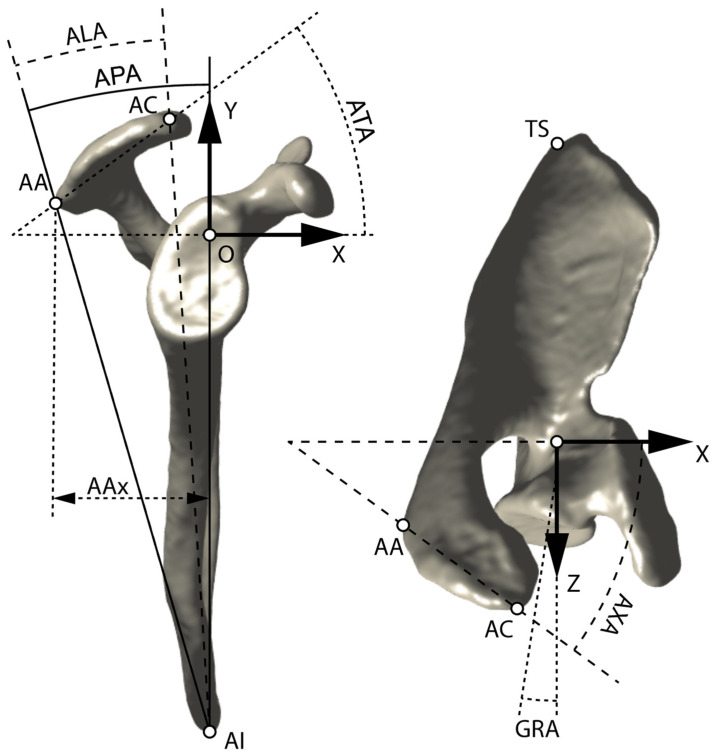
Anatomical description of the scapular coordinate system (OXYZ), acromion landmarks (AA, AC), trigonum spinae (TS), angulus inferior (AI), posterior extension of the acromion (AAx), acromion posterior angle (APA), acromion tilt angle (ATA), acromion length angle (ALA), and glenoid retroversion angle (GRA). The three axes (*x*, *y*, and *z*) correspond to postero-anterior, infero-superior, and medio-lateral, respectively.

**Figure 2 jcm-11-00351-f002:**
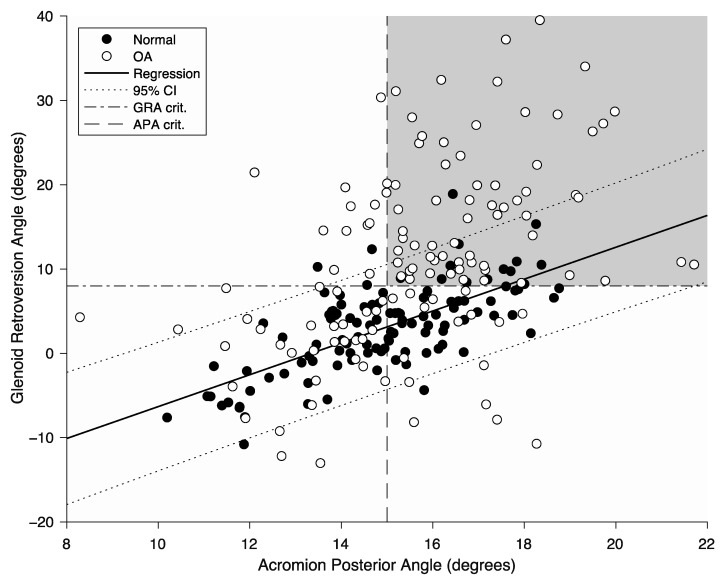
Measured acromion posterior angle (APA, *x* axis) vs. glenoid retroversion angle (GRA, *y* axis) for normal scapulae (black dots) and primary osteoarthritic scapulae (white dots). The continuous line represents the linear regression between the APA and GRA for normal scapulae, with its 95% confidence interval (dotted lines). The grey-shaded area (top right corner) shows the number of osteoarthritic scapulae with critical angle values (dashed lines) of APA > 15 degrees and GRA > 8 degrees.

**Figure 3 jcm-11-00351-f003:**
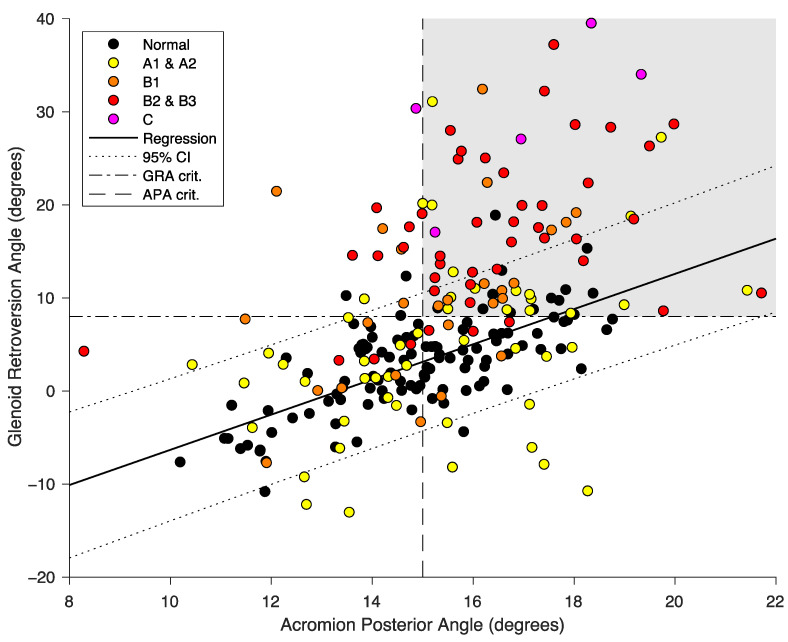
Measured acromion posterior angle (APA, *x* axis) vs. glenoid retroversion angle (GRA, *y* axis) for normal scapulae (black dots) and primary osteoarthritic scapulae subclassified according to the updated Walch classification (yellow, orange, red, and purple dots). The continuous line represents the linear regression between the APA and GRA for normal scapulae, with its 95% confidence interval (dotted lines). The grey-shaded area (top right corner) shows the number of osteoarthritic scapulae with critical angle values (dashed lines) of APA > 15 degrees and GRA > 8 degrees.

**Table 1 jcm-11-00351-t001:** Root mean square errors (RMSE), coefficients of determination (R^2^), and *p*-values of simple and stepwise multiple linear regressions between several acromion landmark coordinates and angles and the glenoid retroversion angle (GRA), for normal scapulae.

	RMSE (Degree)	R^2^	*p*-Value
AAx	3.73	0.480	<0.0001
AAy	5.16	0.006	0.4308
AAz	5.02	0.051	0.0096
ACx	4.31	0.310	<0.0001
Acy	5.14	0.013	0.2298
ACz	5.16	0.007	0.3739
AAx, AAz	3.66	0.505	<0.0001
AAx, AAz, ACx	3.58	0.530	<0.0001
APA	3.73	0.482	<0.0001
ALA	5.17	0.002	0.6305
ATA	5.12	0.022	0.1187
AXA	4.85	0.123	0.0001
APA, ALA	3.61	0.518	<0.0001
APA, ALA, ATA	3.50	0.551	<0.0001

**Table 2 jcm-11-00351-t002:** Glenoid retroversion angle (GRA; mean ± SD) and acromion posterior angle (APA; mean ± SD) for normal and osteoarthritic scapulae, subclassified according to the updated Walch classification.

Scapulae	GRA (Degree)	APA (Degree)
Normal (n = 112)	3.0 ± 5.2	14.9 ± 1.9
Walch type A1–A2 (n = 49)	4.5 ± 9.4	15.3 ± 2.3
Walch type B1 (n = 26)	10.1 ± 9.0	15.2 ± 1.8
Walch type B2–B3 (n = 45)	16.9 ± 8.1	16.3 ± 2.2
Walch type C (n = 5)	29.6 ± 8.4	16.9 ± 1.9

## Data Availability

Authors agree to make data supporting the results or analyses presented in their paper available upon reasonable request.

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
