# Peer review of "Association of the Posterior Acromion Extension with Glenoid Retroversion: A CT Study in Normal and Osteoarthritic Shoulders"

_jcm, 2022, doi:10.3390/jcm11020351_

Round 1
Reviewer 1 Report
The reviewer thanks the authors for their interesting and innovative study. The present study is the first one to analyze the correlation between 3D acromion shape and glenoid retroversion. The potential clinical relevance lies in the determination of risk factors for eccentric glenoid wear. The study does include a normal scapulae group and patients with primary glenohumeral OA. AAx and ACx were identified as factors associated with glenoid retroversion angle.
Overall, the paper is well written and structured. Statistical analyses were done accordingly. Limitations and future approaches were discussed.
What is the main question addressed by the research?
The reviewer thanks the authors for their interesting and innovative study. The study addresses the association between the 3D acromion shape and glenoid retroversion in both normal and osteoarthritic shoulders.
Is it relevant and interesting? / How original is the topic?
The potential clinical relevance lies in the (preoperative) determination of risk factors for eccentric glenoid wear. While scapulae in shoulders with posterior instability were characterized by an increased glenoid retroversion in recent studies, to the best of the reviewer’s knowledge, there are no published data on the correlation between 3D acromion shape and glenoid retroversion. The presented results are thus part of a pilot study addressing these new correlations.
What does it add to the subject area compared with other published material?
This is the first study to determine correlations between 3D acromion shape and glenoid retroversion. Acromion angle (AAx) and anterior point of the acromioclavicular joint (ACx) were identified as factors associated with glenoid retroversion angle. Patient numbers (112 with normal scapulae and 125 patients with primary glenohumeral osteoarthritis) and statistical calculations (multiple linear regression analysis) allowed for sufficient statistical power.
Is the paper well written?
Overall, the paper is well written and structured. To the best of the reviewer’s evaluation, statistical analyses were done accordingly. Limitations and future (clinical) approaches were discussed.
Is the text clear and easy to read?
The manuscript has a clear focus on experienced shoulder consultants with a profound anatomical and surgical background. Understanding of the manuscript thus might be challenging for non-orthopedic surgeons.
Are the conclusions consistent with the evidence and arguments presented?
Results of the present study demonstrated that AAx and ACx were associated with the glenoid retroversion angle. As pointed out by the authors, these angles could be used as a starting point for the determination and measurement of other angles and values such as the acromion posterior angle. This might allow for predictive anatomical parameters or risk factors for posterior glenoid wear in osteoarthritic shoulders using preoperative CT scans in clinical practice.
Do they address the main question posed?
The authors successfully determined possible correlations between 3D acromion shape and glenoid retroversion by first comparing values in normal and then OA shoulders. AAx and ACx were identified as factors associated with glenoid retroversion angle. Further research, including long-term clinical studies addressing functionality and revisions rates might be necessary in the future.
Author Response
Response to reviewers
Reviewer #1
The reviewer thanks the authors for their interesting and innovative study. The present study is the first one to analyze the correlation between 3D acromion shape and glenoid retroversion. The potential clinical relevance lies in the determination of risk factors for eccentric glenoid wear. The study does include a normal scapulae group and patients with primary glenohumeral OA. AAx and ACx were identified as factors associated with glenoid retroversion angle.
Overall, the paper is well written and structured. Statistical analyses were done accordingly. Limitations and future approaches were discussed.
What is the main question addressed by the research?
The reviewer thanks the authors for their interesting and innovative study. The study addresses the association between the 3D acromion shape and glenoid retroversion in both normal and osteoarthritic shoulders.
Is it relevant and interesting? / How original is the topic?
The potential clinical relevance lies in the (preoperative) determination of risk factors for eccentric glenoid wear. While scapulae in shoulders with posterior instability were characterized by an increased glenoid retroversion in recent studies, to the best of the reviewer’s knowledge, there are no published data on the correlation between 3D acromion shape and glenoid retroversion. The presented results are thus part of a pilot study addressing these new correlations.
What does it add to the subject area compared with other published material?
This is the first study to determine correlations between 3D acromion shape and glenoid retroversion. Acromion angle (AAx) and anterior point of the acromioclavicular joint (ACx) were identified as factors associated with glenoid retroversion angle. Patient numbers (112 with normal scapulae and 125 patients with primary glenohumeral osteoarthritis) and statistical calculations (multiple linear regression analysis) allowed for sufficient statistical power.
Is the paper well written?
Overall, the paper is well written and structured. To the best of the reviewer’s evaluation, statistical analyses were done accordingly. Limitations and future (clinical) approaches were discussed.
Is the text clear and easy to read?
The manuscript has a clear focus on experienced shoulder consultants with a profound anatomical and surgical background. Understanding of the manuscript thus might be challenging for non-orthopedic surgeons.
Are the conclusions consistent with the evidence and arguments presented?
Results of the present study demonstrated that AAx and ACx were associated with the glenoid retroversion angle. As pointed out by the authors, these angles could be used as a starting point for the determination and measurement of other angles and values such as the acromion posterior angle. This might allow for predictive anatomical parameters or risk factors for posterior glenoid wear in osteoarthritic shoulders using preoperative CT scans in clinical practice.
Do they address the main question posed?
The authors successfully determined possible correlations between 3D acromion shape and glenoid retroversion by first comparing values in normal and then OA shoulders. AAx and ACx were identified as factors associated with glenoid retroversion angle. Further research, including long-term clinical studies addressing functionality and revisions rates might be necessary in the future.
Author’s response:
Thank you for your appreciation our work. As suggested, we amended the conclusion as follows (conclusion; lines 533-535): “Moreover, long-term clinical studies should evaluate the impact of the APA on clinical function and surgical revision rates after shoulder arthroplasty.” to emphasize the potential role of the APA in helping to predict revision rates and functional outcome after shoulder arthroplasty.

Reviewer 2 Report
The authors present a very well done study looking at the highly complex topic of scapular morphology. They do a nice job presenting results and find some novel measures that correlate to retroversion. This data will add to our understanding of the shoulder and hopefully spur further research on this topic. Specific comments below.
Introduction: Well written
Diagrams: Need More
The authors use trauma pan scans that include the scapula. The cut thicknesses are thick and generally a CT scan of the scapula for diagnostic purposes would have thinner sections. This is evident in the protocol for the CT scans for the OA patients that had thinner thicknesses (.6-1.3mm). How was this accounted for
They use patients aged 18040, but it is known that the ossification centers may not completely fuse until the age of 25, therefore they are mixing partially skeletally mature patients in with skeletally mature patients. The authors state they excluded patients with “open physis.” Open physis of what? Why were patients older than 40 not included?
Why are there more males than females? IT would have been easy to include equal numbers.
The authors included a breakdown by Walch classification but no citation and also no description of what criteria they used for each grade.
More details on the software used fore evaluation. I realize it is linked, but a few sentences describing it would be appropriate.
Include more diagrams, one showing the axis defined would be helpful in addition to just the landmarks.
AAx is not listed on the figure. Again, more diagrams showing the measures on their corresponding axis would be very informative. With so many measures this gets confusing quickly.
How accurate are these measures of the landmarks? Has any inter or intraclass correlations been performed? They seem to have a level of subjectivity to them
“The glenoid centerline was defined by the vector joining 150 the center of the glenoid cavity and the center of a sphere fitting the glenoid cavity” – How was this sphere fit? Was it to the entire glenoid? In arthritic glenoids was this done on the neoglenoid or paleoglenoid
What was the distribution of the measures in the normal?, specifically the GRA. This makes the RMSE values more meaningful
To clarify, a simple statement about the rmeaning of the relationship would be helpful. The positive correlation of the AAx and GRA means that the more posterior it is the more retroversion there is?
The authors present the breakdown of Walch type but then no further analysis is done regarding this and the scapular measures. They should also look at posterior humeral subluxation compared to the 10 parameters they have,
Author Response
Response to Reviewers
Reviewer #2
The authors present a very well done study looking at the highly complex topic of scapular morphology. They do a nice job presenting results and find some novel measures that correlate to retroversion. This data will add to our understanding of the shoulder and hopefully spur further research on this topic. Specific comments below.
Introduction: Well written
Author’s response:
First, we would like to thank the reviewer for her/his appreciation of our work and the insightful comments below.
Diagrams: Need More
Author’s response:
Thank you for this comment. As recommended, we have extended Figure 1 to include all landmarks (including AAx) and angles.
We have further added a new Figure 3 reporting the “subgroup” results for the APA and GRA according to the updated Walch classification.
Figure 3. Measured acromion posterior angle (APA, X-axis) vs. glenoid retroversion angle (GRA, Y-axis) for normal scapulae (black dots) and primary osteoarthritic scapulae subclassified according to the updated Walch classification (yellow, orange, red, and purple dots). The continuous line represents the linear regression between the APA and GRA for normal scapulae, with its 95% confidence interval (dotted lines). The grey-shaded area (top right corner) shows the number of osteoarthritic scapulae with critical angle values (dashed lines) of APA > 15 degrees and GRA > 8 degrees.
Figures are available in the attached word file or in R1 version of the manuscript
The authors use trauma pan scans that include the scapula. The cut thicknesses are thick and generally a CT scan of the scapula for diagnostic purposes would have thinner sections. This is evident in the protocol for the CT scans for the OA patients that had thinner thicknesses (.6-1.3mm). How was this accounted for
Author’s response:
Thank you for this relevant comment. Indeed CT protocols differed between trauma and OA patients. This was already mentioned in the study limitations paragraph (lines 477-481), in the discussion section). “Finally, although trauma patients with normal scapulae and patients with glenohumeral OA were not scanned with the exact same CT protocols, the differences in the reconstructed geometric volumes were small (with slightly smaller voxels for OA than trauma patients) and had no impact on the positioning of scapular landmarks.” In greater detail, the corresponding median voxel size was ~0.80 x 0.80 x 1.25 mm3 and ~0.35 x 0.35 x 0.63 mm3 for trauma CT scans (normal group) and osteoarthritic shoulder CT scans, respectively.
They use patients aged 18-40, but it is known that the ossification centers may not completely fuse until the age of 25, therefore they are mixing partially skeletally mature patients in with skeletally mature patients. The authors state they excluded patients with “open physis.” Open physis of what? Why were patients older than 40 not included?
Author’s response:
Thank you for this insightful comment. We agree that the term “physis” is not appropriate here as it relates to the osteology of the proximal humerus, we therefore corrected towards “ossification center” which is the appropriate term regarding the scapula (Zember JS et al. Radiographics; DOI: 10.1148/rg.2015140254). Regarding the scapula, the ossification centers are multiple and fusion is achieved between 17-25 years of age therefore potential interfering with part of our cohort. As stated in manuscript, CT scans were reviewed and bony fusion confirmed for all patients aged between 18-25 years (n=41 in our normal group). We further looked at difference in glenoid retroversion between the 18-25 years age group and the 26-40 years age group, which was 2.8 + 5.5 degrees and 3.1 + 5.0 degrees, respectively (95%CI -2.2;1.8). Difference regarding APA between the same groups was 14.8 ± 2.1 and 15.0 ± 1.8, respectively (95%CI -1.0; 0.5). Given no significant difference in GRA and APA it appears unlikely that the correlation between retroversion with APA is affected in these subgroups. We modified the method accordingly (Materials and methods, lines 90-91): “Exclusion criteria were any radiological sign or history in medical records of disorders of the shoulder bones and joints (OA, fracture, glenoid dysplasia, or prior surgery of the upper limb), CT signs of immature skeleton (absence of fusion between any of the scapular ossification centers [16]) or CT artifacts (motion or metal).”
Regarding the age limit of 40 years old for the “normal group”, this limit was set arbitrarily but with the goal to avoid including scapulae/patients with slight degenerative changes. As these changes appear progressively over time it makes inclusion/exclusion challenging especially in the age group 40-60 years old. While some patients have clear signs of osteorarthritis, others have just slight joint space narrowing, and/or subtle osteophytes and/or subchondral sclerosis which might be overlooked and therefore falsely create a continuum between normal and osteoarthritic shoulders. We however agree with you that the “perfect” control group should be matched in terms of age. We therefore checked that this demographic parameter did not affect the critical APA value in the osteoarthritic group as already mentioned in the study limitation (Discussion, lines 476-477).
Why are there more males than females? IT would have been easy to include equal numbers.
Author’s response:
Thank you for this valuable comment. As per study design, the “normal group” was created out of our trauma patient database. Trauma being more common in males and shoulder osteoarthritis more common in females it led to this altered distribution of gender. We agree with you, that this factor could potentially influence our results. Our aim was to assess potential morphological associations separately, first in normal and then in osteoarthritic scapulae. As stated in the study limitation paragraph (Discussion, lines 470-476), we verified that the same correlations held true in males and females, with variations within the 95% CIs. We are therefore confident that this limitation did not affect interpretability of the presented results.
The authors included a breakdown by Walch classification but no citation and also no description of what criteria they used for each grade.
Author’s response:
Thank you for this relevant comment. We used the updated Walch classification. We used the modification of the original Walch as published by Bercik et al. (JSES 2016; DOI: 10.1016/j.jse.2016.03.010). The citation was added to method section (Line 116). FB who is the senior musculoskeletal radiologist and co-author of the study classified all glenoids according to the updated Walch classification on three-dimensional CT reconstructions. The same criterias as in the citation were used (Bercik et al, JSES 2016; DOI:10.1016/j.jse.2016.03.010). We therefore used the updated Walch classification defining a A-type as a centered humeral head with central glenoid erosion, in the A2 glenoid pattern the line connecting the concavity transects the humeral head (10.1016/j.jse.2016.03.010). The B-type has a posterior humeral head subluxation with the B2 presenting an articular biconcavity and the B3 glenoid defined with 15 degrees or more of glenoid retroversion. Lastly, C-type glenoids are defined as dysplastic glenoids with at least 25 degrees of retroversion regardless of erosion. We modified the method accordingly (Method section, Lines 117-119).
More details on the software used fore evaluation. I realize it is linked, but a few sentences describing it would be appropriate.
Author’s response:
Thank you for this comment, on lines 176-177 (original manuscript), we extended
“The glenoid centerline and all other 3D quantities and computations defined above were performed with Matlab (MathWorks).”
as follows (Materials and Methods R1 manuscript, lines 184-187)
“The glenoid centerline and all other 3D quantities and computations defined above were performed with a custom-made code written in Matlab (MathWorks). This script takes as input the three coordinates of all the scapular landmarks, and all the points on the surface of the glenoid cavity to provide all reported measurements in the scapular coordinate system, for each case individually [19].”
Include more diagrams, one showing the axis defined would be helpful in addition to just the landmarks.
Author’s response:
Thank you for this comment. The three axes X, Y, and Z, corresponding to postero-anterior, infero-superior, and medio-lateral, respectively, are all represented in Figure 1. To clarify this point, we have extended the legend of Figure 1 (Lines 149-150).
AAx is not listed on the figure. Again, more diagrams showing the measures on their corresponding axis would be very informative. With so many measures this gets confusing quickly.
Author’s response:
Thank you for this comment. AAx is the x component of the landmark AA. We have added AAx on Figure 1 to clarify this point.
How accurate are these measures of the landmarks? Has any inter or intraclass correlations been performed? They seem to have a level of subjectivity to them
Author’s response:
Thank you for this valuable comment. We did not perform ICC for landmark positions for this study. This is therefore a study limitation which is specifically addressed in the discussion (Discussion, lines 461-470). “The major limitation of our method was the manual identification of the two acromion landmarks, the effect of which was minimized by the good to excellent reliability in the positioning of scapular landmarks by a single experienced human observer (Terrier et al. Measurements of Three-Dimensional Glenoid Erosion When Planning the Prosthetic Replacement of Osteoarthritic Shoulders. The Bone & Joint Journal 2014, DOI: 10.1302/0301-620X.96B4.32641). This could be improved by using sophisticated fully automatic landmark detection methods (Taghizadeh, E et al. Automated CT Bone Segmentation Using Statistical Shape Modelling and Local Template Matching. Computer Methods in Biomechanics and Biomedical Engineering 2019, DOI:10.1080/10255842.2019.1661391). However, even with such automated methods, the AC might still be affected by osteoarthritis, conversely to all other anatomical landmarks used to characterize the acromion morphology, the angles, and the local (scapular) coordinate system. Nevertheless, the AC is not related to the APA or the scapular coordinate system, and its variability caused by OA is supposedly weak since we found no significant difference when comparing the normal and osteoarthritic datasets.”
“The glenoid centerline was defined by the vector joining the center of the glenoid cavity and the center of a sphere fitting the glenoid cavity” – How was this sphere fit? Was it to the entire glenoid? In arthritic glenoids was this done on the neoglenoid or paleoglenoid
Author’s response:
Thank you for this comment. The sphere was fit on the entire glenoid cavity, manually identified by the change of curvature at the rim. For arthritic glenoids, we considered the entire glenoid surface (neoglenoid and paleoglenoid) too. The method was already published in 2014 with citation in the manuscript (Terrier, A.; Ston, J.; Larrea, X.; Farron, A. Measurements of Three-Dimensional Glenoid Erosion When Planning the Prosthetic Replacement of Osteoarthritic Shoulders. The Bone & Joint Journal 2014, 96-B, 513–518, doi:10.1302/0301- 404 620X.96B4.32641.).
What was the distribution of the measures in the normal?, specifically the GRA. This makes the RMSE values more meaningful
Author’s response:
Thank you for your comment. The distribution was normal as already mentioned on lines 260-261 of the result section. “Data for the six acromion landmarks, four acromion angles, and the GRA all followed a normal distribution.”
To clarify, a simple statement about the meaning of the relationship would be helpful. The positive correlation of the AAx and GRA means that the more posterior it is the more retroversion there is?
Author’s response:
Thank you for this comment. We agree this formulation might potentially be misleading. We modified the sentence as follow (Lines 242-244 of results section): “For osteoarthritic scapulae, we also observed a significant positive correlation between the APA and GRA (R2 = 0.197, p < 0.0001), suggesting that a higher APA was associated with an increased GRA (Figure 2).”
The authors present the breakdown of Walch type but then no further analysis is done regarding this and the scapular measures. They should also look at posterior humeral subluxation compared to the 10 parameters they have,
Author’s response:
Thank you for this valuable comment. We added a “Figure 3” to better visualize the effect of the Walch classification on the correlation between glenoid retroversion and APA.
Figure 3. Measured acromion posterior angle (APA, X-axis) vs. glenoid retroversion angle (GRA, Y-axis) for normal scapulae (black dots) and primary osteoarthritic scapulae subclassified according to the updated Walch classification (yellow, orange, red, and purple dots). The continuous line represents the linear regression between the APA and GRA for normal scapulae, with its 95% confidence interval (dotted lines). The grey-shaded area (top right corner) shows the number of osteoarthritic scapulae with critical angle values (dashed lines) of APA > 15 degrees and GRA > 8 degrees.
Figures are available in the attached word file or in R1 version of the manuscript
While by definition glenoid retroversion increase between type A, B and C glenoids, this trend is not seen regarding APA (Figure 2 and Table 2). A possible hypothesis is that a large APA especially above 15 degrees is related to increased glenoid wear appearing over time. We modified results section as follow (lines 244-246): “While both the GRA and APA increased with the (alphabetical) progression of the updated Walch class, the increase in APA was not proportional to that of the GRA (Table 2, Figure 3).”
We finally added a table 2 to present the influence of the Walch classification on the correlation between the GRA and APA. Given the small numbers in each subgroup (even we tried to pool A, B and C types) limiting interpretability and statistical analysis.
Table 2. Glenoid retroversion angle (GRA; mean ± SD) and acromion posterior angle (APA; mean ± SD) for normal and osteoarthritic scapulae, subclassified according to the updated Walch classification.
Scapulae |
GRA (degree) |
APA (degree) |
Normal (n=112) |
3.0 ± 5.2 |
14.9 ± 1.9 |
Walch type A1-A2 (n=49) |
4.5 ± 9.4 |
15.3 ± 2.3 |
Walch type B1 (n=26) |
10.1 ± 9.0 |
15.2 ± 1.8 |
Walch type B2-B3 (n=45) |
16.9 ± 8.1 |
16.3 ± 2.2 |
Walch type C (n=5) |
29.6 ± 8.4 |
16.9 ± 1.9 |
Lastly, this work focused on the link between three-dimensional acromion and glenoid morphology. Posterior humeral head subluxation was not directly assessed in this work as it was not its focus.

Reviewer 3 Report
This is a comparative analysis of CT scans of normal and osteoarthritic shoulders analyzing the association between acromion morphology and glenoid version. There is no clear study question and no hypothesis is stated in this paper. The aim was to provide a better understanding of the etiology of eccentric glenoid wear. In my opinion this study does not provide any explanation for eccentric glenoid wear since only the scapula morphology was examined without any relation to the humeral head or soft tissue (rotator cuff, capsule, ligaments…) or pathology. A correlation was found between posterior acromion extension and glenoid version angle in asymptomatic shoulders. This correlation was weak in patients undergoing arthroplasty for any reason. No information is provided about this second group of symptomatic patients and many questions arise from this fact: How was the glenoid version in this cases, how was the glenoid wear, was there static posterior subluxation or even patients with anterior wear?
The methodology is well described and performed. Nevertheless, the definition of glenoid version or the here presented glenoid retroversion angle (GRA) is confusing and does not meet the commonly recognized glenoid version among shoulder surgeons according to the Friedman method. This methodological issue has to at least be discussed in the method section since this definition of GRA is focusing more on the glenoid vault than the orientation of the scapula and with it the medial border.
I recognize the authors’ effort to study a barely understood clinical problem but I can not see any additional information this study can provide to understand posterior shoulder instability or the etiology of osteoarthritis with or without posterior eccentric wear. A more detailed discussion about the clinical problems and the presented results can support your thesis.
Author Response
Response to Reviewers
Reviewer #3
This is a comparative analysis of CT scans of normal and osteoarthritic shoulders analyzing the association between acromion morphology and glenoid version. There is no clear study question and no hypothesis is stated in this paper. The aim was to provide a better understanding of the etiology of eccentric glenoid wear. In my opinion this study does not provide any explanation for eccentric glenoid wear since only the scapula morphology was examined without any relation to the humeral head or soft tissue (rotator cuff, capsule, ligaments…) or pathology. A correlation was found between posterior acromion extension and glenoid version angle in asymptomatic shoulders. This correlation was weak in patients undergoing arthroplasty for any reason. No information is provided about this second group of symptomatic patients and many questions arise from this fact: How was the glenoid version in this cases, how was the glenoid wear, was there static posterior subluxation or even patients with anterior wear?
Author’s response:
Thank you for these comments regarding our manuscript. The objective of our study was to evaluate the association between three-dimensional acromion morphology and glenoid retroversion. This work being designed as a pilot study, it was not possible to have a hypothesis other than the presence or absence of the aforementioned correlation.
This work is to the best of our knowledge the first to report a positive correlation between posterior acromion extension and glenoid retroversion. We agree that this does not provide a pathophysiologic explanation at this stage. However, starting from this observation, it allows to explore possible hypothesis including related soft tissues (e.g. deltoid orientation, influence of humeral head position).
This work focused on the link between three-dimensional acromion and glenoid morphology. Posterior humeral head subluxation was not directly assessed in this work as it was not its focus. We however provide a sub analysis in the osteoarthritic group between the updated Walch classification and APA.
The methodology is well described and performed. Nevertheless, the definition of glenoid version or the here presented glenoid retroversion angle (GRA) is confusing and does not meet the commonly recognized glenoid version among shoulder surgeons according to the Friedman method. This methodological issue has to at least be discussed in the method section since this definition of GRA is focusing more on the glenoid vault than the orientation of the scapula and with it the medial border.
Author’s response:
Thank you for this valuable comment. We are not completely sure to understand to the full extent of your comment. However, the three-dimensional method used to determine glenoid retroversion was previously published (Terrier, A.; Ston, J.; Larrea, X.; Farron, A. Measurements of Three-Dimensional Glenoid Erosion When Planning the Prosthetic Replacement of Osteoarthritic Shoulders. The Bone & Joint Journal 2014, 96-B, 513–518, doi:10.1302/0301- 404 620X.96B4.32641.). Regarding version, the aforementioned study reported inter- and intra-observer reliability of 0.96 and 0.83, respectively. It seems therefore a reliable and valid method to determine version in 3D. We modified method as follow (Lines 181-182):” This method has previously shown good to excellent inter- and intra-observer reliability [19]”
I recognize the authors’ effort to study a barely understood clinical problem but I can not see any additional information this study can provide to understand posterior shoulder instability or the etiology of osteoarthritis with or without posterior eccentric wear. A more detailed discussion about the clinical problems and the presented results can support your thesis.
Author’s response:
Thank you for this comment. The main objective of this work was to assess the correlation between acromion shape and glenoid retroversion. To the best of our knowledge this was not previously reported. We address the clinical importance and summarize current knowledge in this field in the second and third paragraph of the Introduction:
“Over the past three years, Beeler et al. analyzed shoulder computed tomography (CT) images to better characterize the acromion roof morphology. Compared with osteoarthritic shoulders, these authors reported that the acromion was more externally rotated (axial plane), more downward tilted (coronal plane), and had wider posterior coverage of the glenoid (sagittal plane) in shoulders with rotator cuff tears [7]. The same research group further found a significant difference between shoulders with concentric and eccentric primary glenohumeral OA, and concluded that a flatter acromion roof with less posterior glenoid coverage could contribute to static posterior subluxation of the humeral head and posterior glenoid wear [8]. Furthermore, Meyer et al. were able to link a de-creased posterior acromion slope and increased glenoid retroversion with static posterior subluxation of the humeral head and posterior glenoid wear [9]. More recently, Beeler et al. reported that the scapula of shoulders with dynamic and static posterior instability was characterized by an increased glenoid retroversion and an acromion that is shorter posterolaterally, higher and more horizontal in the sagittal plane [10].
Static posterior subluxation of the humeral head and posterior glenoid wear are of particular interest to shoulder surgeons, as they have been related to both early glenohumeral OA in young adults [11], and higher complication rates after shoulder arthroplasty [12]. Although Walch et al. stated that static posterior subluxation of the humeral head preceded posterior glenoid wear, with glenoid retroversion as a risk factor [11,13], there is currently no consensus regarding this chicken-or-egg debate [6,14,15]. To our knowledge and despite the recent study by Beeler et al. [10], there are no published data on the correlation between the detailed three-dimensional (3D) acromion shape and glenoid retroversion.”
At this stage we can solely hypothesize the influence of acromion morphology on muscle forces. We discuss in detail the relationship between our work and previously published papers as well as potential outlooks as follows:
“When secondarily looking at osteoarthritic scapulae, the correlation between AAx and the APA was still significant but weaker than in normal scapulae (R2 = 0.482, p < 0.0001 vs. R2 = 0.197, p < 0.0001, respectively). This translates that the correlation observed in normal scapulae seems to be disrupted in primary glenohumeral OA patients. Our hypothesis is that this might be related to posterior glenoid wear. Previous research identified scapulae with increased glenoid retroversion and/or posterior glenoid wear as a risk factor for implant failure in total shoulder arthroplasty [18,19]. In addition, posterior glenoid bone loss is known to progress over a 5- to 15-year timeframe in up to 55% of patients [20]. Our research might therefore be critical to help council patients by defining a critical APA value related to posterior glenoid wear. We first identified a critical GRA with a ROC curve analysis. Then, by using the Youden index, we determined the optimal cut-off value of the APA that could distinguish between scapulae above and below this critical GRA threshold.
Glenoid retroversion was also correlated with ACx, but ACx was strongly correlated with AAx (see correlation matrix in Supplementary material), meaning that the relative AP acromion length (distance between AA and AC) has a low variability. Glenoid retroversion was also negatively correlated with the lateral extension of the AA (AAz), which was further partly correlated with the posterior extension (Table 1). These correlations between the two acromion landmark coordinates were also present between the four tested acromion angles, and explain why they do not all appear in the multiple corre-lations obtained with the stepwise multiple linear regression analysis.
It appears likely that the strong correlation observed between the acromion and glenoid in normal scapulae is determined by the end of growth [21]. While several hypotheses regarding posterior glenoid wear have been raised, including premorbid glenoid retroversion [22], muscular imbalance [23], and lower humeral retroversion [24], its pathophysiology remains unknown. We might reasonably assume that the acromion affects the glenoid through the action of muscles, but this remains purely conjectural. This link might also be more deeply anchored in the evolutionary process of homo sapiens [25].
Normalized values of the two important acromion landmarks that are the AA and AC joint have not been previously reported. However, a wide range of acromion angles have recently been described with increasing interest in characterizing the acromion morphology. The APA presents similarities with the “posterior glenoid coverage” pro-posed by Beeler et al., as both are based on AAx and the medio-lateral scapular plane [7,8]. However, Beeler et al. used the glenoid center as the middle point, while we used the AI instead. Not only because the AI is not affected by glenohumeral osteoarthritis, in contrast to the glenoid center, but also and primarily to better correlate the APA with the posterior extension of the acromion (AAx). These two angles are thus very different, and we verified that the “posterior glenoid coverage” was not correlated with the GRA in our series of normal scapulae.
The ALA corresponds to the distance between AA and AC in the sagittal (XY) plane, measured as an angle using AI as the third landmark. Again, it is similar to the “overall glenoid coverage” proposed by Beeler et al. [7,8], but uses AI instead of the glenoid center as the middle point, to better correlate with the AA-AC segment (R2 = 0.908 vs. R2 = 0.093, respectively). In our series of normal scapulae, these two angles were only weakly correlated (R2 = 0.127), and the variability range of ALA was five times lower.
The ATA corresponds approximately to the previously defined acromion tilt,[26] or 90 degrees minus the posterior acromion slope [9], or the sagittal tilt.[7,8] The ATA was 25.2 ± 8.2 (range, 5.2-46.9) degrees in our normal scapulae vs. 23.4 ± 8.7 (range, 4.5-42.5) degrees in osteoarthritic scapulae, which corresponds closely to the values in the articles referenced above.
The AXA corresponds approximately to the previously defined axial tilt angle [7,8]. The AXA was 26.1 ± 8.9 (range, 4.2-52.1) degrees in our normal scapulae vs. 29.1 ± 9.9 (range, 2.5-52.1) in osteoarthritic scapulae, which also matches the previous works mentioned above.

Round 2
Reviewer 3 Report
Thank you for your comments!
No more suggestions.